# *In Vitro* Models of Diabetes: Focus on Diabetic Retinopathy

**DOI:** 10.3390/cells13221864

**Published:** 2024-11-11

**Authors:** Giulia Galgani, Giorgia Bray, Alma Martelli, Vincenzo Calderone, Valentina Citi

**Affiliations:** 1Department of Pharmacy, University of Pisa, 56126 Pisa, Italy; giulia.galgani@phd.unipi.it (G.G.); alma.martelli@unipi.it (A.M.); vincenzo.calderone@unipi.it (V.C.); valentina.citi@unipi.it (V.C.); 2Interuniversity Centre for the Promotion of the 3R Principles in Teaching and Research, Italy

**Keywords:** diabetic retinopathy, alternative models, *in vitro* models, cell cultures, organoids

## Abstract

Diabetic retinopathy is a major eye complication in patients with diabetes mellitus, and it is the leading cause of blindness and visual impairment in the world. Chronic hyperglycemia induces endothelial damage with consequent vascular lesions, resulting in global vasculitis, which affects the small vessels of the retina. These vascular lesions cause ischemic conditions in certain areas of the retina, with a consequent increase in the release of pro-angiogenic mediators. In addition to pharmacological interventions for controlling the blood glycaemic level, the main strategies for treating diabetic retinopathy are the intravitreal injections of drugs, surgical treatments, and vitrectomies. The complexity of diabetic retinopathy is due to its close interactions with different cell types (endothelial cells, astrocytes, and Müller cells). The evaluation of the efficacy of novel pharmacological strategies is mainly performed through *in vivo* models. However, the use of different animal species leads to heterogenic results and ethical concerns. For these reasons, the development of new and reliable *in vitro* models, such as cell co-cultures and eye organoids, represents an urgent need in this area of research. This review features an overview of the *in vitro* models used to date and highlights the advances in technology used to study this pathology.

## 1. Introduction

Diabetic retinopathy (DR) is one of the main ocular complications of diabetes that leads to blindness and visual impairment. The amount of people suffering from DR, according to the International Diabetes Federation, is expected to increase rapidly in a few years by more than 50% [1,2].

Briefly, high glucose (HG) levels alter retinal homeostasis due to a massive increase in glucose metabolites that promotes both the generation of reactive oxygen species (ROS) and an increase proinflammatory mediators. These processes mainly damage the endothelial cells (ECs) of small retinal vessels, arterioles, capillaries, and venules, triggering ischemic events in certain areas of the retina. In response to the ischemic events, pro-angiogenic factors are released, including vascular endothelial growth factor (VEGF), which induces neovascularisation that causes a decrease in retinal transparency and an increase in vascular permeability, leading to the development of retinal edema [3,4,5].

DR is generally a non-symptomatic condition in the early stages, but as the disease progresses, hypercoagulability, thickening of the membrane, loss of pericytes (PCs), and reduced blood supply lead to the disruption of the blood–retinal barrier (BRB). In addition, neurodegeneration of neural cells in the retina, which can precede vascular apoptosis, leads to the complete loss of vision at the final stage of the disease [6,7].

As it is a complex disease that involves numerous cell types, the most common strategy for investigating the pathophysiological processes of DR involves using animal models, including mice, rats, and dogs [8]. In these *in vivo* models, DR can be induced by pharmacological treatment, such as with streptozocin (STZ), which is able to destroy pancreatic β-cells and provoke the onset of type I diabetes [9,10], or by a high-sugar or high-fat diet; these treatments can be coupled together or with streptozocin [11]. Other methods for obtaining *in vivo* DR have been reviewed by Olivares and coworkers [8] and include the intravenous injection of alloxan, the first drug recognized as a diabetes inducer, as it can destroy Langerhans islets. In addition, surgical intervention by removing the pancreas completely or partially (only β-cells) causes a dramatic increase in glucose plasma levels and a rapid onset of DR. Since DR is characterized by neovascularization, this condition is obtained by exposing juvenile animals to a hyperoxic environment, which results in hypoxia when animals are transferred to normal conditions or undergo cytokine treatment. Genetic models are also used and can be divided into animal groups with spontaneously arising diabetes or genetically modified diabetes. In particular, db/db, non-obese diabetic (NOD) mice, and BioBreeding (BB) rats are spontaneously arising diabetic animal models [12,13] that are widely used in research because they do not need a pharmacological treatment or a high-sugar/high-fat diet to develop the pathology. They are of great importance for research in metabolic disorders, but their chronic pathological state leads to major suffering conditions that healthy animals do not experience. These animals develop several comorbidities, such as infertility [13], depression, psychosis-like symptoms [14], and kidney complications, which can influence the physiological response to pharmacological treatments [15].

Although the models previously described are useful tools for elucidating the pathophysiology of DR, they exhibit several limitations: the overall complexity of human DR is not fully represented because eye structures anatomically vary from species to species, with pigs and zebrafish being the two species with eyes most similar to human eyes [8]; however, their breeding habits and lifespans are completely different.

For these reasons and due to ethical concerns about animal use and suffering, the development of *in vitro* models for reproducing DR disease is ongoing and ranges from “basic” models (monolayer cell cultures) to more complex systems, including co-cultures with two or more cell lines, tissues, or organ-on-a-chip systems.

In particular, DR *in vitro* models are based on cultures of human retinal cell lines that can be used with human or non-human glial cells. Monolayer cell cultures are the easiest tool for understanding cell responses to specific damages, drug mechanisms of action, and toxicity. Single-cell lines can also be cultured in combination, creating *in vitro* co-cultures that reproduce cell-to-cell signaling, protection, and interactions, helping to define the pathological pathways of the disease. The most advanced *in vitro* models are represented by 3D models, organ-on-a-chip models, and organoids that can mimic not only cell-to-cell signaling but also retinal structures and architectures, giving a better overview of its functioning.

The aim of this review is to describe the structure of the retina, focusing on how the heterogenous cells that comprise this organ can communicate under hyperglycemic conditions. This manuscript serves as a basis for enhancing the understanding of the different *in vitro* models that have been developed until now, starting from monolayer models to more complex 3D models.

## 2. Structure of the Retina

The retina is an almost transparent tissue located on the back of the eyeball over the choroid. It is divided into two parts: the neuroretina is composed of nine layers (inner limiting membrane, nerve fiber layer, ganglion cell layer, inner plexiform layer, inner nuclear layer, outer plexiform layer, outer nuclear layer, external limiting membrane, and photoreceptor layer) and the retinal pigmented epithelium (RPE), which is separated from the choroid by Bruch’s membrane [16,17] (Figure 1).

The conversion of light stimulus into images is achieved by the interactions of different neuronal cells sited in the various layers of the retina transmitting signals to the brain cortex. Photoreceptors are the first cells involved in this pathway, and they are divided into cones and rods that contain two proteins, called respectively opsin and rhodopsin, which are activated by light. When these proteins are activated, cells undergo membrane-graded changes, and the release of neurotransmitters occurs, generating the signal. Photoreceptors have bodies in the photoreceptor cell layer; cones are responsible for the vision of colors, while rods are responsible for the vision of shapes [18,19]. The signal is then transmitted to bipolar cells through the modulation of other neurons, called horizontal cells, that collect and regulate information by several photoreceptors. Both bipolar and horizontal cells have bodies located in the inner nuclear layer, and the interactions of these two types of cells with photoreceptors take place in the outer plexiform layer [20]. Bipolar cells play an important role in transmitting signals from the outer to the inner part of the retina, from photoreceptors to ganglion cells. For this reason, they make synapses with all the other retinal neuronal cells and are 180 degrees orientated, interacting with other cells in the outer plexiform layer, where they reach horizontal cells and photoreceptors, as previously described, and in the inner plexiform layer, where they relate to ganglion cells and amacrine cells [21]. Amacrine cells, which also have their bodies in the inner nuclear layer, give temporal order to the information that goes from bipolar cells to ganglion cells by synapses among the three cell types in the inner plexiform layer [22]. Ganglion cells, whose bodies are located in the ganglion cell layer, receive a signal that has been collected and filtered, and they are responsible for the transmission of the visual stimulus to the brain cortex, where an image is created [22,23].

Along with neuronal cells, three types of glial cells are located in the retina: astrocytes and Müller cells (MCs), which form the macroglia, and microglial cells. Their role is to maintain retinal homeostasis and prevent neuronal damage, with some differences: MCs provide nutrients to neuronal cells, discard neuronal waste, recycle neurotransmitters (in particular, glutamate), and modulate immune responses; astrocytes are mainly involved in maintaining BRB integrity and neurotrophic support; and microglial cells are immune sentinels that regulate apoptosis, acting mainly as phagocytes of dead cells.

MCs are the main component of retinal glia. Their somas are in the nuclear layer, and their main process is to spread from the inner to the outer limiting membrane, giving structural stability to the entire retinal tissue [22,24,25]. Astrocytes are located near blood vessels, and they are almost absent where thin nerve fiber layers reside [26]. They express VEGF, inducing angiogenesis, and their proliferation is promoted by a ligand named platelet-derived growth factor receptor alpha (PDGFRA), which is secreted by retinal ganglion cells [27]. Microglial cells are spread all over the retina and represent the primary immune cell type [28] α, which is secreted by retinal ganglion cells [27]. Microglial cells are spread all over the retina and represent the primary immune cell type [28].

### 2.1. Retinal Vascularisation

Blood vessels that reach retinal cells are disposed in layers and can be divided into two main categories: retinal vessels, which provide nutrients to the neuroretina by macroglia cells and are distributed at both sides of the inner nuclear layer, and choroidal microcirculatory system, which bring nutrients to the outer part of the retina (in particular, RPE) [27,29]. These two capillary systems originate from different independent vascular beds: retinal vessels come from both the superior and inferior temporal arteries, and their angiogenesis is promoted by astrocytes, expressing VEGF; the choroidal system originates from a branch of the ophthalmic artery: the short posterior ciliary artery [27]. The flux of substances, ions, proteins, and liquids from vessels to the retina is regulated by BRB and established by tight junctions (TJs) of neighboring ECs (inner BRB) and pigmented epithelial cells (outer BRB), preventing retinal damage from liquid extravasation; it alters the chemical environment and reduces substances permeability as well. Its function is sustained by glial cells, such as MCs and astrocytes, with processes in touch with BRB and are cells that play a pivotal role in maintaining its integrity [30].

### 2.2. Structure and Role of BRB

BRB, whose main role is maintaining normal visual function [31], is constituted by two distinct components: the inner BRB (iBRB) and the outer BRB (oBRB) [31].

The iBRB regulates the exchange of solutes between the retinal vascular lumen and the neural retina. It consists of specialized retinal vascular ECs characterized by TJs, which limit non-specific transport in the vascular circulation. The ECs are surrounded by PCs, incorporated into a basement membrane, and in communication with glial cells, such as MCs and astrocytes, contributing to their structural and functional integrity [32,33,34,35].

The oBRB consists of a fenestrated choroidal vascular network adjacent to a monolayer of RPE that is separated by a thin layer of extracellular matrix (Bruch’s membrane). RPE is important as it controls the flux of nutrients between the choroidal capillaries and the photoreceptor layer. Under pathological conditions, the vascular growth of the choroid invades Bruch’s membrane and induces the break of the RPE [31].

BRB plays several roles, including the maintenance of osmotic pressure, electrolyte balance, and tissue hydration through the regulation of the flow of essential molecules, including ions, proteins, and water, between the systemic circulation and the retina. Furthermore, the retina is a tissue with a high energy demand, and it is mainly glucose-dependent. For this reason, BRB promotes the transport of glucose from the blood to the retinal neurons via specialized glucose transporters. Additional functions of BRB include the elimination of metabolic waste products, such as CO_2_ and nitrogen metabolites, from the retina into the circulation by preventing the accumulation of toxic substances. Furthermore, BRB protects the retina from inflammatory damage and autoimmune reactions by limiting the entry of immune cells and inflammatory molecules from the circulation [35].

## 3. Development of DR

DR has been considered a microvascular pathology for many years. However, recent studies have demonstrated that HG-induced neuronal damage precedes vascular dysfunction, making DR a neurovascular disease [36]. This section aims to describe the molecular basis of the pathophysiological pathways resulting in the development of DR and describes how these modifications induce pathological alterations in different types of cells (Figure 2).

### 3.1. Hyperglycemia-Related Pathophysiological Pathways

Hyperglycemia triggers several pathological pathways that promote neuronal and vascular damage, such as the deposition of advanced glycation end products (AGEs), the activation of protein kinase C (PKC), and the enhanced polyol pathway, which results in high levels of intracellular ROS. Therefore, RPE and activated glial cells release pro-inflammatory mediators, including tumor necrosis factor α (TNF-α), interleukin (IL) 1-β, IL-6, chemokines, and adhesion molecules. This is the beginning of the inflammatory response to the hyperglycemic stimulus that results in the activation of nuclear factor—kappa B (NF-κB). NF-κB translocates in the nucleus and activates the transcription of inflammatory mediators and adhesion molecules, including intercellular adhesion molecule-1 (ICAM-1) and vascular cell adhesion molecule-1 (VCAM-1), that attract monocytes and leukocytes, causing neuronal and vascular damage [37,38,39].

#### 3.1.1. Deposition of Advanced Glycation End Products

The development and accumulation of AGEs are pathological pathways that may be related to the onset of DR. In hyperglycemic conditions, their generation is particularly intensified due to the presence of an elevated concentration of glucose [37]. In diabetic patients, AGEs are constantly detected in retinal vessels, and their concentrations correlate with the severity of retinopathy [38].

AGEs include various molecules deriving from the “Maillard reaction”, which is a glycation reaction involving primary amines of macromolecules (lipids, proteins, and nucleic acids). Their formation begins with the generation of Schiff base adducts that spontaneously rearrange into the relatively stable Amadori products [39,40].

The crosslinks between AGEs and macromolecules alter the structures and functions of the proteins, resulting in a decrease in the elasticity, consequently increasing the thickness and rigidity of the vessel wall.

Furthermore, AGEs interact with the cell surface AGE-binding receptors (RAGEs) galectin-3 and CD36, regulating multiple cellular pathways, including the NF-κB pathway, which promotes the transcription of crucial genes, such as growth factors, cytokines, IL-6, IL-1β, TNF-α, and adhesion molecules (endothelin-1, VCAM-1, ICAM-1) [41,42,43].

Diabetic patients show elevated levels of carbohydrate-derived oxidation products, which can be considered evidence of diabetes-related oxidative protein damage [44].

AGEs accumulate in RPs, activating the apoptotic process and leading to cell loss [45]. In addition, AGEs promote the translocation of NF-κB to the nucleus, especially in PCs in hyperglycemic conditions, stimulating ROS production [46]. PCs cover all the retinal vessels and maintain microvascular homeostasis and integrity [47]. Thus, PC loss due to the accumulation of AGEs can be considered one of the earliest morphological changes in DR.

Besides PC loss, the deposition of AGEs promotes the dysfunction of the basement membrane, resulting in the hyperpermeability of retinal capillaries and the disruption of endothelial junctions, leading to BRB disruption, which occurs at an early stage.

#### 3.1.2. Activation of Protein Kinase C

PKC regulates signal transduction pathways that are triggered by specific stimuli [6]. As a final event, the increase in cytosolic diacylglycerol (DAG) occurs, resulting in the synthesis of the classical -α, -β, and -δ isoforms of PKC [48]. Hyperglycemia is related to high levels of DAG, leading to the formation of glycerol-3-phosphate, which subsequently activates a vicious cycle and promotes the *de novo* synthesis of DAG. In patients with diabetes, vascular tissues and retina show high amounts of DAG. Furthermore, *in vitro* evidence using human retinal PCs (hRPs) demonstrates a direct correlation between glucose and DAG levels [49]. As a final event, DAG leads to retinal vascular dysfunction by altering enzymatic activities in the endothelium and in retinal PCs (RPs). Notably, the PCK-β isoform drives the expression of VEGF, which controls vascular permeability and promotes angiogenesis [50].

#### 3.1.3. Enhancement of the Polyol Pathway

Even though the polyol pathway is a marginal pathway of glucose metabolism, it is believed to play a key role in the onset and development of DR [51]. The polyol pathway becomes active when glucose intracellular levels are elevated [52]. In this manner, glucose is metabolized through the enzyme aldose reductase (AR) expressed in the retina, leading to the formation of sorbitol while consuming nicotinamide adenine dinucleotide phosphate (NADPH) as a cofactor. Sorbitol is then transformed into fructose by sorbitol dehydrogenase (SDH). Because sorbitol is unable to cross cell membranes, it accumulates in cells and undergoes slow metabolism to fructose. Sorbitol accumulation in retinal cells has multiple detrimental effects, including osmotic damage and AGE production. In addition, consuming NADPH reduces the activity of glutathione reductase, which is responsible for the endogenous production of reduced glutathione and is important for the protective response against oxidative stress [6].

#### 3.1.4. Increase in Oxidative Stress

In diabetes, oxidative stress is caused by several factors, including the decrease in the number of antioxidant molecules, the activation of the polyol pathway, and changing redox balances.

The retina is remarkably exposed to oxidative stress since it has a high polyunsaturated acid content. It is characterized by oxygen uptake and glucose oxidation. In addition, both the antioxidant defense enzymes responsible for the elimination of ROS and reactive nitrogen species (RNS), the number of non-enzymatic antioxidant molecules, such as vitamin C, vitamin E, and β-carotene, are decreased during hyperglycemia-induced oxidative stress [53]. In this way, excessive amounts of ROS are responsible for the oxidation of biomolecules, such as DNA, proteins, carbohydrates, and lipids [54].

#### 3.1.5. Inflammatory Response

Hyperglycemia induces the abnormal functioning of mitochondria, increases the regulation of ROS production, damages vascular ECs, induces cell death, and increases inflammatory factors [55]. Inflammation promotes significant alterations of molecular pathways associated with the onset of DR, particularly in the early stages of the disease.

The previously analyzed pathological mechanisms certainly underlie and sustain the inflammatory process due to the increase in ROS production. The inflammatory response itself triggers a vicious cycle that sustains oxidative stress, polyol pathways, and the accumulation of AGEs through the production of cytokines and adhesion molecules, the overexpression of RAGEs, the increase in the nitric oxide (NO) level, and NF-κB factor signaling activation.

Subclinical inflammation in the retina leads to the formation of new weak vessels, and their increased permeability due to VEGF leads to hemorrhaging in the retina and leukostasis [56]. Leukostasis is a crucial event in DR leading to the occlusion of arterioles and endothelial cell death, which in turn locally amplifies the inflammatory response in retinal tissue [56]. Patients with diabetic retinopathy show elevated levels of several inflammatory cytokines, such as IL-1β, IL-6, IL-8, TNF-α, and monocyte chemoattractant protein-1 (MCP-1), highlighting the role of inflammation in the early stages of DR and the progression of the inflammatory response in all retinal cell types [57].

### 3.2. Cell Response to Hyperglicaemic Condition

#### 3.2.1. Vascular Damage

ECs play an important role in maintaining the homeostasis of retinal blood vessels. They represent the inner layer of retinal vessels and are implied in the release of vasoactive mediators, including NO [33].

Endothelial dysfunction begins with the deposition of AGEs that directly trigger the inactivation of eNOS, reducing the level of NO release in retinal vessels. NO is a gasotransmitter that helps to maintain vascular tone and elasticity and is implied in antioxidant mechanisms; thus, its depletion leads to vascular thickening and oxidative stress [57].

Furthermore, the interactions of AGEs with their RAGE receptors activate pathways related to the increase in oxidative stress, including the activation of NADPH oxidases and NF-κB translocation. Moreover, hyperglycemia reduces the activity of the peroxisome proliferator-activated receptor-γ (PPAR-γ), which physiologically inhibits NF-κB, promoting the progression of retinal inflammation [58].

Once activated, NADPH oxidases interact with PKC, leading to the increase in the levels of cytosolic DAG, ROS, and RNS, thereby resulting in the activation of inflammatory processes and EC dysregulation [6,7,34,57,59].

As ECs are involved in the structure of iBRB and are surrounded by glial cells and PCs that help provide nutrients and restore altered conditions, their dysfunction leads to iBRB impairment. iBRB plays a fundamental role in preventing vascular leaks and regulates vessel permeability; EC death and the destruction of their TJs cause the loss of their function, resulting in leaks, bleeding, macular oedemas, and retinal ischemia [3,4,6,60].

Consequently, to restore retinal homeostasis, ECs release pro-angiogenic factors, including VEGF. VEGF is a growth factor released not only by ECs after the interaction of AGEs with their specific receptors but also by glial cells, PCs surrounding blood capillaries, and RPEs. VEGF mainly triggers the formation of new collateral vessels after an ischemic event and leads to vasodilation, oedemas, and macular degeneration, causing vision problems in the last stages of DR [4,61,62,63].

#### 3.2.2. Loss of Pericytes

PCs are the first and last vascular elements of the retina’s neurovascular unit (NVU) that control functional hyperemia. Their contractile processes ensheathe capillaries, promoting vasodilation and vasoconstriction under specific stimuli, while their connection with the other types of cells provides the NVU with precise signals of the source of neuronal activity [64]. This initial response can spread along neighboring PCs and ECs to control blood flow tens of micrometers away from the site of stimulation [64,65]. The propagative nature of PC activity plays a critical role in both local network interactions and in signaling to regulate NVU retinal blood flow upstream in precapillary arterioles. While the very definition of PCs has been contested with respect to their contractile abilities [66], investigations of the vasomotor response to electrical stimulation have revealed three distinct vasoactive propagation modes [65]. Types A, B, and AB correspond to a jumping spread (node mode), a gradual spread (tide mode), and an undulatory spread (wave mode), respectively.

Chronic hyperglycemia has been reported to markedly affect pericytes and their connectivity with other cells. In a diabetic model induced by streptozotocin administered in rats that developed DR, the PC and EC connectivity levels were reduced by 70% [67]. Furthermore, the density of PCs was reduced by 15%, with a consequent thinning of the pericyte that wrapped capillaries [65]. Advanced diabetes caused a complete loss of PCs [68,69]. Diabetes-induced changes to pericyte function appeared much earlier, affecting their roles in both capillary diameter control and vasomotor response signaling along the vascular tree.

#### 3.2.3. Glucose Effect on MCs and Astrocytes

Macroglia is essential in regulating retinal homeostasis, affecting both neuronal and vascular cells. MCs are the most important glial cells in the retina because their form and position let them contact every other retinal cell type. In healthy patients, they mainly provide structural stability, with their axons and dendrites being inserted into all the layers; deliver nutrients to the other cell types by regulating electrolytes, removing metabolism products and toxins, and recycling neurotransmitters; and participate in the neuronal immune system [25,70]. Astrocytes are mainly involved in providing nutrients, releasing trophic factors, removing toxic agents from vascular cells, reducing oxidative stress, and modulating inflammation [33,70].

The alteration of glucose levels and the consequent inflammatory response lead to the activation of macroglia, which changes its morphology and activity [71]. Astrocytes decrease in number and functioning, thus aggravating the impairment of blood vessels that lose their first homeostatic support and cannot prevent BRB destruction; MCs experience morphological modifications and hypertrophy, leading to changes in their structural support, losing their ability to remove toxic agents from the environment and starting to release pro-inflammatory mediators and metalloproteinases (MMP) that are involved in TJ degradation [25,30,70].

Therefore, due to the missing structural support physiologically assured by macroglia, retinal homeostasis is compromised, causing the degeneration of neuronal cells and photoreceptors. The final involvement of photoreceptors leads to the production of superoxide and inflammatory proteins at this level, causing vision loss [72]. The release of pro-inflammatory cytokines promotes the overexpression of inducible NO synthase (iNOS), with a consequent release of NO. NO and activated macroglia play an important role in vascular perfusion, as they are involved in a process called neurovascular coupling. Neurovascular coupling is a multifactorial response to altered neuronal activity that leads to augmented blood flow, retinal venous and arterial dilatation, vascular leaks, and macular edema [73,74,75,76,77]. The presence of NO itself leads to the formation of RNS, which can cause cell death and aggravate oxidative stress [78].

### 3.3. BRB Impairment

BRB plays an important role in preventing vascular leaks, regulating the passage of nutrients and substances, and avoiding the leakage of toxic agents into the environment. Therefore, its impairment is the main process that leads to DR-induced visual alterations. As previously stated, BRB is composed of two parts: iBRB and oBRB. The first part is composed of neighboring ECs that are in close contact by means of TJs, such as occludins and claudins; the second is composed of RPE cells linked by TJs and adherent junctions [79]. The endothelium of the iBRB is coupled with glial cells and PCs to form NVU. NVU plays the important role of reducing oxidative stress and inflammation of the iBRB and phagocyte waste products to maintain barrier homeostasis [33]. High levels of glucose modifying the functionality of glial cells lead to reductions in all these repairing and safeguarding processes; moreover, inflammation and oxidative stress are not controlled, and pro-inflammatory cytokines are released, causing vascular endothelium damage and iBRB destruction [80]. This is the first mechanism of vision impairment because of DR; after this, the activation of MMPs and oxidant agents affects RPE in the last stages of the pathology, leading to oBRB dysfunction and complete blindness [81].

BRB disruption represents a pivotal element in the pathogenesis of numerous retinal pathologies, including DR. In DR, chronic hyperglycemia results in damage to retinal ECs, impairing the function of the iBRB and leading to increased vascular permeability and the formation of microaneurysms. The depletion of PCs is a key feature in the pathogenesis of DR that represents a critical step in the breakdown of the BRB. This damage can result in retinal edema that is characterized by the accumulation of fluid in the extracellular space of the retina, which may lead to decreased vision and, in severe cases, permanent visual impairment in diabetics [82].

## 4. *In Vitro* Retinal Models

*In vitro* models of DR are valuable tools for understanding the mechanisms involved in this pathology, and they vary from monolayer cell cultures to co-cultures, 3D models, and organoids (Figure 3). Cell monocultures are the first and easiest way to understand the roles of different agents, either protective or harmful, for a single cell type, in addition to understanding the mechanisms used to maintain cellular homeostasis. They are usually less expensive than other models and guarantee high reproducibility and repeatability. A combination of multiple cell cultures in co-cultures helps the understanding of their interaction, showing how their collaboration can exacerbate harm or sustain the healing process. Although these models are employed in the early stage of a disease study, they inevitably lack the complexity of the entire animal model. For this reason, animals are still used in DR studies. However, ethical concerns and the impossibility of having a single type of animal eye that completely represents the human one led to the development of new *in vitro* models, such as 3D models and organoids that are structurally similar to human physiological conditions, thus representing a potentially useful tool for creating a comprehensive overview of the mechanisms involved in the pathology. The development of these models is rapidly increasing, although the difficulty of recreating an entire microsystem is the reason for the limited number of these models on the market.

### 4.1. Cell Cultures

Retinal cell cultures are valuable tools for understanding the pathophysiological mechanisms that regulate the development of DR. Cell models may be classified according to their composition and structure. The simplest are monolayer cell cultures, which comprehend either neuronal cells, glial cells, or PCs. Co-cultures are more complex models, which include ECs. The section below describes the *in vitro* models available for the evaluation of the effects of the HG condition on the cell response, starting from monolayer cell cultures to more complex *in vitro* systems (Table 1).

#### 4.1.1. Monolayer Cell Cultures

Monolayer cell cultures usually consist of RPE and glial cells. The most used RPE cell is ARPE-19, which spontaneously arises in human RPE cells and is widely used in retinal and epithelium functional studies [98,99]. They maintain high differentiation, epithelium properties, and the ability to be depolarized under an appropriate stimulus; these characteristics are useful for studies on BRB integrity, including evaluations of TJs and cell viability [100]. Several biomarkers of DR can be monitored with this model, like apoptotic factors caspase-3 and 9, peroxidases, and VEGF release [101]. Other RPE cells used in monolayer cell cultures are fetal human RPE cells (fhRPE), which are primary cells that show better barrier properties compared to ARPE-19 [98,99,102].

Glial cells are another essential component of retinal wellness and homeostasis because they provide nutrients, give structural stability, and regulate immunity defence. Glial cell models of MCs, astrocytes, and microglia are commonly used for the evaluation of their response and activation under hyperglycemic and stressing stimuli as models of DR. Cell viability, physiology, and pathological biomarkers can be assessed, monitoring the changes in the release of cytokines and oxidative stress under HG conditions [83,84,103,104].

Tu and coworkers used MCs derived from the eyes of 3-day-old newborn C57BL/6 mouse pups to study the role of geniposide, an antioxidant agent, in reducing hyperglycemia-derived oxidative stress. As chronic hyperglycemia induces the activation of MCs, promoting glycolysis and the release of pro-inflammatory cytokines that can cause the death of retinal cells, in particular retinal ganglion cells, they aimed at demonstrating the role of geniposide in attenuating DR-related MC activation [103]. Zong and coworkers used MC cultures (MIO-M1) to evaluate the upregulation of RAGE and S100B (RAGE ligand) in diabetic retina. MCs were cultured under HG conditions (25 mmol/L), demonstrating that RAGE and its signalling pathway are upregulated in a hyperglycemic environment [83].

BV-2 are mouse microglial cells that are widely used *in vitro* for studying microglial responses under different pathological conditions. Some studies [84,85] use BV-2 cells to evaluate the role of hypoxia in pathological mechanisms, maintaining them in a hypoxic environment with 1% O_2_ or adding them to the culture medium CoCl_2_ for 8 h. While Xie aimed to evaluate the role of the Kir 2.1 channel in trigeminal neuralgia [85], Jiang used these cells to study the role of fractalkine, in preventing DR [84]. They demonstrated that hypoxia could activate microglia and promote neuroinflammation and is involved in the upregulating pathways of microglia apoptosis.

A novel method for studying retinal diseases is based on the development of models derived from stem cells. When cultured under an appropriate stimulus, stem cells have the ability to differentiate into different retinal cells widely used in regenerative therapy [105,106]. They are mainly employed as precursors of photoreceptors and RPE cells, and they are valuable tools for creating complex models ranging from co-cultures to 3D structures and organoids of retinal tissue.

The most used stem cells are human embryonic stem cells (hESCs) and human induced pluripotent stem cells (hiPSCs). hESCs, derived from embryos, are pluripotent cells that can be physiologically differentiated from all specialized cells; their ability to be converted into different cell types is still under investigation because optimal culture conditions have to be assessed [107]. hiPSCs are differentiated cells reconverted to a pluripotent state after exposure to appropriate factors expressed either by oocytes or by pluripotent cells. They regain the ability to differentiate the primary germ layers (ectoderm, endoderm, and mesoderm), and they have a developmental potential similar to that of hESCs [108]. Changing the culture conditions of hiPSC and hESC leads to RPE cells and photoreceptors [109,110,111].

In particular, Rowland and coworkers differentiated hESC line H9 and iPSC (iPS(IMR90)-3) to RPE using purified extracellular matrix proteins to obtain an efficient and optimized protocol that can overcome differences among cell donors. Cells were cultured with Dulbecco’s modified Eagle’s medium (DMEM)/F12/GlutaMAX I medium supplemented with knockout serum replacement (20%), non-essential amino acids (NEAAs) (1×), β-mercaptoethanol (0.1 mM), and the addition of basic fibroblast growth factor (bFGF; 4 ng/mL) for hESCs, and zebrafish bFGF (zbFGF; 100 ng/mL) for iPSCs. For the differentiation, zbFGF or bFGF was removed and the cells were seeded onto mouse embryonic fibroblasts (MEFs). This procedure lasted about 80 days until the pigmented cells could be enriched with pigmented spots. After this first passage, pigmented cells were removed and reseeded with RPE medium composed of DMEM (HG), F12 (30%), and B27 (2%) for further investigations, such as the assessment of the role of the extracellular matrix on cell differentiation [110].

Furthermore, Mellough and coworkers have studied a method to develop photoreceptors using hESC and/or hiPSC for mimicking neuronal and retinal development. The abilities of hESC and hiPSC to regulate the expression of a range of factors are important for retinal cell type specification for neural and retinal phenotypes. The efficient generation of photoreceptors was obtained in 45 days. hESC and hiPSC were cultured under atmospheric oxygen conditions on mitotically inactivated MEFs in hESC medium containing DMEM, 1 mM L-glutamine, 100 mM of NEAA, 20% knockout serum replacement, 1% penicillin–streptomycin and 8 ng/mL bFGF. hESC/hiPSC medium was changed daily and hESC/hiPSC colonies were passaged every 4–5 days at a ratio in the range of 1:3–1:4 by incubation in 1 mg/mL collagenase IV. To differentiate cells into photoreceptors, culture medium was supplemented with growth factors, Activin A, Shh, and T3 in three steps. This method could be important for future studies of cell replacement in people suffering from retinal damage and blindness [112].

As previously stated, methodologies have been established for the generation of photoreceptors and RPE from hiPSCs. This approach is straightforward and offers significant advantages for investigating the pathogenesis of DR, as RPE cells derived from hiPSCs exhibit morphological, physiological, and functional attributes comparable to those of human RPE [113]. The same models were then used by Mellough and colleagues with minor modifications to create 3D-laminated neural retina with the addition of insulin-like growth factor 1 (IGF-1) to obtain a more complex structure of neural tissue and RPE [114] that was later used for investigations on the stages of retinal development and RNA transcription during retinal maturation [115]. However, although these models are promising for future applications in the treatment of DR-related photoreceptor loss and neuronal damage, no further investigations on this topic for these models have been carried out yet.

For *in vitro* experiments, RPs are generally used as co-cultures with hREC for BRB investigations to evaluate how the two cell types interact in healthy and pathological situations [87,116]. Several studies have used adipose stem cells (ASCs) as monocultures differentiated into PCs. In 2013, Mendel and coworkers demonstrated that ASCs can be differentiated into PCs by performing morphological analysis, studying the location of intravitreal injection, and verifying the presence of PC marker expressions, such as nerve/glial antigen 2 (NG2), PDGFR-β, and alpha smooth muscle actin. The differentiation was conducted from purified human ASCs (hASCs) and murine ASC (mASCs), which were cultured on culture dishes at 37 °C, 5% of CO_2_, and 75% of relative humidity, using cells from passage 3 to 6. The differentiation medium included DMEM/F12 with 10% of fetal bovine serum (FBS) and 1% of antibiotic/antimycotic solution. Further studies included intravitreal injections of differentiated PCs to analyze whether these cells were able to reach a perivascular location and to maintain PC’s protective role; the aim was to use them as remedies for retinal diseases [88]. The same culture conditions of mASCs were used by Cronk and colleagues to investigate whether ASCs derived from healthy patients and diabetic patients maintained the same ability to protect retinal microvessels, showing that there was a decrease in their function when derived from unhealthy donors [89]. Periasamy differentiated hASCs into PCs to examine the role of PDGFR-β, a PC marker expressed on ASCs, showing that it had positive effects on the endothelial angiogenic properties [90], while Mannino and coworkers investigated different culture media to assess which one led to the closest PC cell type by comparing these cultures to human RPs and understanding that the best culture condition implied the use of the PC culture medium. Moreover, they elucidated the ability of hASCs to protect hREC cellular junctions [91]. Then, Mannino used this model to investigate the protective role of ASCs on hRECs by showing that HG levels did not impact the ASC proliferative pattern or migration ability, and, when co-cultured with hREC, they could reduce ROS production and increase the expression levels of anti-inflammatory markers, such as IL-10; by transforming growth factor-β1 (TGF-β1) [92], they showed the protective properties toward BRB that could be studied for autologous administration on patients [93].

Other studies demonstrated that hESC could be differentiated into cells that are functionally and phenotypically close to PCs. These hESC derived cells are called hESC-PVPCs [97].

#### 4.1.2. Co-Cultures

The combination of multiple cell cultures yields a good overview of the mechanisms underlying DR pathogenesis, showing interactions and signaling pathways that can occur between different types of cells (Table 1).

#### 4.1.3. Endothelium and Pericytes

Co-culture models with PCs and ECs are designed to mimic the microvascular environment and BRB to study the interactions between these two cell types.

Tarallo and colleagues have developed two co-culture models for the study of DR using human retinal ECs (HRECs) and primary hRPs. In one model, HRECs are grown on a porous membrane suspended in the culture medium, while HRPs are grown at the bottom of the same well. This makes it possible to study the effects of soluble factors released by HRECs on hRPs in a hyperglycemic environment, simulating the *in vivo* situation in which endothelium is the first target of hyperglycemia. In the second one, HRECs and hRPs are grown on opposite sides of the same porous membrane, allowing both direct contact between the cells and interaction via soluble factors. This model makes it possible to study the effects of both direct contact and soluble factors on cell proliferation, apoptosis, and tubule formation. The results obtained with this co-culture model provide valuable information on the cellular mechanisms underlying DR and highlight the importance of the interactions between HRECs and hRPs in the pathogenesis of the disease. The model also offers a platform to test potential therapies for DR [86].

Fresca and colleagues constructed an *in vitro* model of the BRB, a three-dimensional system that aims to replicate the physiological environment of the human BRB. HRECs, hRPs, and human retinal astrocytes were used. The three cell lines were co-cultured in a ratio of 1:5:5 for PCs, ECs, and astrocytes, respectively. The tri-culture system was constructed using Transwell inserts and 12-well plates. Both Transwell inserts and 12-well plates were coated with poly-L-lysine (PLL) to promote cell adhesion. PCs were seeded on the underside of the Transwell inserts and allowed to adhere for 4 h. The inserts were then inverted and inserted into the 12-well plates containing pericyte culture medium. Astrocytes were seeded on the bottom of the 12-well plates precoated with PLL and allowed to adhere overnight. The next day, Transwell inserts containing PCs were placed in the 12-well plates with the astrocytes. The ECs were then seeded on the upper side of the Transwell inserts above the PCs. The *in vitro* model of BRB was formed within three days after the start of co-culture. To simulate the hyperglycemic conditions of diabetes, glucose was added to the culture medium until a final concentration of 40 mM was reached. The cells were exposed to this condition for 48 h. This detailed protocol illustrated the meticulous construction of the *in vitro* model of BRB, allowing researchers to study the effects of HG on cellular interactions within BRB and to evaluate potential therapies for DR [87].

#### 4.1.4. Microglial and Endothelial Cells

Retinal vascular alterations are one of the main reasons for visual impairment in patients affected by DR. Although several studies focus their attention on the role of endothelial growth factor in this pathology, some mechanisms still remain unclear and may involve glial cells, whose role is to maintain retinal and vascular homeostasis. For this reason, a co-culture involving HMC-3 cells, human microglial cells, and PUMC-HUVEC-T1 (Peking Union Medical College-human umbilical vein ECs T1) was used by Wu and coworkers for recreating microglial-ECs interaction under HG conditions, which is typical of DR. The aim of this model was to understand the mechanisms involved in vascular regeneration after the hyperglycemic attack and the role of microglial cells in maintaining vascular homeostasis [94].

#### 4.1.5. Retinal Pigmented Epithelial Cells and Endothelial Cells

BRB is composed of two parts: the inner part is characterized by TJs between vascular ECs, while the outer part has TJs between RPE cells. Oliveira and coworkers studied a new strategy to develop a 3D cellular model of the BRB by co-culturing these two cell types using ARPE-19 for RPE and HUVEC for ECs. This co-culture was maintained inside two different layers of gels: the inner part over ARPE-19 cells was composed of hydrogels created to mimic human vitreous, while vitrified collagen gels were used as substitutes for Bruch’s membrane and were in close contact with HUVEC cells. This model aimed to reproduce the barrier properties typical of BRB that had a particular impact when intravitreal injections were performed. ARPE-19 cells were grown on monolayers to replicate the retinal pigmented epithelium. At first, the cells were cultured separately under appropriate conditions; then, ARPE-19 and HUVEC were placed in close contact; finally, HUVEC were grown on a biomaterial membrane that simulated Bruch’s membrane, while the vitreous humor was modeled on the monolayer of ARPE-19 cells using a hydrogel made of hyaluronic acid and collagen [95].

#### 4.1.6. MCs and Microglia

The role of the inflammatory pathway in retinal diseases was evaluated in the study conducted by Xin Hu and coworkers to understand whether the inflammatory state was influenced by the interactions of different glial cells. In this study, MCs and microglia were cultured on Transwell inserts. MCs were activated with the addition of 3,5-dihydroxyphenylglycine (DHPG), and the interactions with the other cell types were evaluated. The study demonstrated that the activation of macroglia seemed to consequently activate microglia, exacerbating the retinal inflammatory state [96] (Table 1).

### 4.2. 3D Models

As previously stated, DR is a complex condition, and the development of a human-like *in vitro* model of this disease still remains a major challenge for researchers. Two-dimensional models are the most used *in vitro* models, and they still offer advantages in high-throughput screenings. However, 2D models only allow us to delve into the pharmacological response of a single cell type. Animal models have been useful for elucidating the progression and the characterization of the disease in great detail and have helped to develop appropriate therapeutic options [8]. However, these models still have ethical limitations, and the results are often inconsistent with human pathophysiology. In order to overcome the challenges and restrictions associated with using *in vitro* and animal models, scientists have recently developed 3D models that can be exploited as further tools in pharmacological studies. Indeed, merging results from 2D and 3D models offers a wide view of the potential mechanisms involved in the pathogenesis of complex diseases (Table 2).

3D models are often derived from human cells and are developed to reproduce the complex interactions among the different types of cells involved in the onset and progression of DR [129].

#### 4.2.1. Retinal Tissue

Several studies aim at optimizing the use of pluripotent stem cells from mice and humans for the generation of retinal tissue that closely resembles native tissue. Alavi and Baranov employed eye shells containing a range of retinal cells, including ganglion cells, rod and cone photoreceptors, amacrine and bipolar cells, and MCs [130].

Furthermore, McUsic and colleagues developed a tissue-level retinal model employing dissociated cells, which can be applied in a few techniques, including drug screening and regenerative medicine [117]. They examined the construction of retinal tissues from dissociated mouse retinal cells and cultures derived from hESCs. The study focused on the use of scaffolds designed to mimic the anatomical arrangement of neurons and glia in the mammalian retina, promoting cell survival and the development of advanced tissue properties. The researchers fabricated scaffolds with 15-μm-diameter channels based on the size of radial columnar units in the rodent retina. RPE was obtained from postnatal mouse retinal cells of the Nrl-GFP strain cultured on poly (lactic-co-glycolic acid) microchannels. The processes of Müller’s glia and rod photoreceptors were oriented parallel to the walls of the microchannels. HESC-derived retinal cells cultured in the scaffolds exhibited even more distinct stratification than mouse cells, suggesting the potential of scaffolds to promote tissue differentiation and organization. Co-cultures of scaffolds with retinal cells and RPE promoted the increased maturation of photoreceptors. Using dissociated cells placed on a scaffold that guided cell orientation, McUsic et al. recreated an ordered retinal tissue that could be applied in a few situations, including drug screening and regenerative medicine.

Furthermore, Zhong and coworkers showed how to differentiate retinal tissue from iPSCs for the generation of 3D cell aggregates that spontaneously develop eye-like domains. These domains, enriched in retinal precursors, can be collected and grown in suspension to form 3D retinal cups. These retinal cups show layer-specific stratification and give rise to all major retinal cell types arranged in their appropriate layers. The 3D retinal cups show features similar to the human embryonic retina, including a polarized and pseudostratified epithelium with proliferating cells in interkinetic nuclear migration. Neural retinal cells differentiate spontaneously, following the characteristic wave of neurogenesis from the center to the periphery and migrating into the corresponding retinal layers. This pattern of differentiation is similar to that of the vertebrate retina [118].

However, despite the potential use of these retinal-like structures for further investigations on retinal diseases, no further studies have been conducted on these models thus far.

#### 4.2.2. Organoids

3D tissues can be recreated using stem cells that mimic the characteristics and functions of a real organ. These structures are known as organoids [131].

HiPSC and hESCs cells are the most used cell lines for the development of retinal organoids, which represent valuable models for the evaluation of the impact of the HG condition and for the investigation of new potential pharmacological treatments.

Eiraku and coworkers investigated the potential use of mouse ESCs (mESCs) for the construction of organoids. They demonstrated that mESCs form hemispherical vesicles in an autonomous manner when cultured under appropriate conditions with a polarized neuroepithelium, which subsequently develops into laminated retinal structures [132].

Cowan and coworkers created human retinal organoids with two synaptic and three nuclear layers using hiPSCs [119]. Their basic structure is similar to the human retina, and they are often made of spheres with layers of cells inside. This model can be exploited to identify new cellular targets, promote the healing of human retinas, and understand the complex pathophysiological pathways in DR.

Another organoid was created by Wimmer and coworkers to investigate diabetic vasculopathy. A self-organising 3D human blood vessel organoid from hiPSC and hESC-H9 was described. This approach ensures the faithful reproduction of the characteristics and functionality of human microvasculature. Cells were differentiated using DMEM, F12 media, 20% KnockOut Serum Replacement (KOSR), glutamax, and NEAA. Cell aggregates were exposed to 12 μM of the inhibitor of glycogen synthase kinase-3α Laduviglusib (CHIR99021) on day three; then, bone morphogenetic protein 4 (BMP4), VEGF-A, and fibroblast growth factor 2 (FGF-2) were added on days five, seven, and nine. Day 11 showed that the cells moved to a medium containing 10 μM of a potent inhibitor of activin receptor-like kinase 5 named SB43152, 30 ng/mL of FGF-2, and 30 ng/mL of VEGF-A to prevent excessive pericyte differentiation and boost endothelial yield. ECs and PCs found in these human blood vessel organoids self-assembled into capillary networks encased in a basement membrane. They were cultivated with 75 mM glucose to replicate the diabetes condition, while non-diabetic control conditions contained just 17 mM glucose [120].

Magranè and coworkers developed a method for creating an organoid resembling the human retina by cultivating hiPSCs in repeated cultures. The cells were differentiated on Matrigel using DMEM/F12 supplemented with 5% FBS, 0.1 mM NEAA, 2 mM GlutaMax, 1% N2, 1% B27, 10 mM β-glycerol phosphate, recombinant human IGF1 (10 ng/mL), and 10 mM nicotinamide supplemented with Noggin (10 ng/mL), Dickkopf-1 DKK1 (10 ng/mL), and bFGF (10 ng/mL). The medium was changed every other day for 30 days. The multizone ocular progenitor cells (mzOPCs) in the cell culture on day 30 were responsible for the formation of the ocular-specific structures named the surface ectoderm, neural crest, lens, stromal cells, NR, and RPE. On day 30, mzOPCs were picked to generate multiocular organoids, cultured in the same medium, and added to retinoic acid for 60 days. On day 90, the RPE and retinal organoids were separated and cultured in a medium supplemented with triiodothyronine (T3, 20 nM) to allow for photoreceptor maturation and RPE pigmentation for 140 days. At the end, an RPE cell monolayer was obtained on Matrigel plates. The different ocular organoids created in this work have potential use in drug testing, disease modeling, developmental biology, and customized medicine [121].

#### 4.2.3. Spheroids

Spheroids are cells that can self-aggregate and form spherical structures. Multicellular spheroids can be used to represent redifferentiation, re-epithelialization, and Bruch’s membrane formation [133]. This technique has the potential to reconstitute the biological signaling pathways of cell–cell interactions that support cell proliferation and viability, hence resolving issues with monolayer cultures, such as the restricted realization of multicellular microenvironments *in vivo* [134].

Sato and coworkers created a spheroid culture from human RPE cells to understand the role of Bruch’s membrane and lipoproteins in the onset of DR. Furthermore, this model can also be exploited for studying eye aging and the pathogenesis of RPE-related diseases. The cells were first conventionally cultured in culture plates and then resuspended in a culture medium containing 20% stock methylcellulose solution (v/v) for creating spheroids. After 7 days, the spheroids were transferred and cultured in non-adherent culture dishes [123].

Nam and colleagues designed an oBRB–choriocapillary model based on a microphysiological system to mimic the architecture and function of the oBRB *in vivo*. First, to mimic the choriocapillary, a 3D microvascular network was created in the central channel of the device. This network was created by seeding HUVEC and normal human lung fibroblasts (NHLF) in the central channel. Then, it was suspended in a fibrinogen and thrombin solution. They studied the function of fibroblasts in neo-angiogenesis and found that fibroblasts helped limit the random proliferation of endothelial cells and regulate blood vessel generation. ARPE-19 cells were added to one of the side channels to form the oBRB. The medium was added with VEGF to promote the formation of a microvascular network. To confirm the physiological structure of their oBRB–choriocapillary model, they detected the polarization and expression characteristics of the TJs of the RPE, Bruch’s membrane, and the fenestrated diaphragm of the choriocapillaries. Finally, they reproduced the diabetes mellitus environment, which promoted a decrease in TJ integrity, loss of the barrier function, and narrowing of the blood vessels, comparable to what happens in human pathological conditions, replicating retinal and choroidal diseases *in vitro* for drug testing [124].

De Lemos and coworkers created an *in vitro* 3D model of DR using hiPSC-derived retinal organoids. The model focuses on diabetic retinal neurodegeneration (DRN), an early event in DR characterized by neuronal cell death in the retina. Then, 100-day-old retinal organoids were exposed to different concentrations of glucose to mimic the hyperglycemic environment of DR. After 6 days of glucose treatment, retinal organoids were analyzed for various aspects of DRN, including neurodegeneration, which was assessed by quantification of pycnotic nuclei (a hallmark of cell death) and staining with Fluoro-Jade C, which identifies degenerating neurons. The loss of specific retinal cells was analyzed by immunofluorescence to determine changes in the number of different retinal cell types, including retinal ganglion cells, amacrine cells, and photoreceptor progenitors. Inflammation was examined by assessing morphological changes in MCs using vimentin staining and by measuring the expression and secretion levels of pro-inflammatory markers, such as VEGF, IL-1β, and MCP-1. Oxidative stress was determined by measuring intracellular levels of ROS using the DCF-DA probe and analyzing the expression and protein levels of antioxidant enzymes. Using this model, the researchers were able to demonstrate that high glucose treatment induces a number of changes in retinal organoids consistent with the initial DRN, including increased cell death, loss of retinal ganglion cells and amacrine cells, glial reactivity, inflammation, and oxidative stress. These results demonstrate the utility of this 3D model to study the mechanisms of DRN and to identify potential therapeutic targets [122].

## 5. Organ-on-a-Chip

Organs-on-a-chip are more trustworthy than 2D culture tissues since they have a structure more similar to an *in vivo* tissue. The innovative feature of organs-on-a-chip is that different configurations can be obtained depending on the tissue to be reconstructed. They consist of glass, plastic, or polydimethylsiloxane (PDMS) microchannels into which cells are seeded. This compartmentalization can facilitate the connection between different cell types (Table 2) [32].

In order to replicate cell–cell interactions in a constrained area, traditional 3D models have been combined with microfluidic techniques. Using human retinal microvascular ECs and choroidal fibroblasts, which self-assemble into a three-dimensional perfusable network, Paek et al. employed a microfluidic technology to reproduce the vascular structure. RPE cells created from hiPSCs are layered over this network, forming a barrier, generating basement membrane proteins that resemble Bruch’s membrane, and exhibiting pigmentation. This configuration creates a system that reproduces the outer BRB, generating [98] an accurate *in vitro* model of vascularized 3D tissues that mimic the natural process of *de novo* blood vessel development. This model can be used to predict drug toxicity, develop new therapeutic strategies, and understand therapeutic mechanisms of action and the mechanistic understanding of the disease [32,125].

To further enhance the comprehension of the disease, the researchers employed the construction of organoids using co-cultured neural retina and RPE explants, thereby facilitating more efficacious interactions between photoreceptors and RPE cells. To safeguard the structural integrity of retinal organoids, Achberger and colleagues co-cultured retina-on-a-chip derived from hiPSCs embedded in a hyaluronic acid-based hydrogel on a layer of human-RPE PSCs in a vasculature-like perfusion model [127].

Yeste and coworkers used HREC, a human neuroblastoma cell line (SH-SY5Y), and ARPE-19 to recreate the BRB. A new microfluidic system was created, and cells were interconnected between a grid of grooves to facilitate their contact. HREC was used to set up the inner BRB, SH-SY5Y to set up the neuroretina, and ARPE-19 cells for the outer BRB. SH-SY5Y, HREC, and ARPE-19 cells were seeded in different compartments of a microfluidic device coated with fibronectin. The device with fibronectin was rinsed with a culture medium and incubated for 30 min, obtaining a microfluidic cell culture system to emulate tissue barrier properties [126].

Maurissen and colleagues created an iBRB-on-a-chip model using a microphysiological system for vascular modeling that consists of small, interconnected channels and chambers that mimic the structure and function of blood vessels. The microfluidic devices include a central channel for culturing cells in a fibrin gel and two side channels flanking the central channel. The side channels allow the perfusion of culture media and other factors, mimicking nutrient supply and waste removal in living tissue. To achieve this, three primary cell types of the human retinal microvasculature—human retinal microvascular endothelial cells (hRMVECs), hRPs, and human retinal astrocytes (hRAs)—were mixed in a 1:1:1 ratio in a fibrin gel. To initiate the formation of the microvascular network, the culture medium was supplemented with a high concentration (50 ng/mL) of VEGF. After four days, the VEGF concentration was reduced to a basal level (5 ng/mL) to promote quiescence and induce barrier properties. Using a microfluidic system, the researchers were able to create a 3D environment in which primary retinal cells could self-assemble into a network of functional vessels. The precise control of growth factors such as VEGF allowed the researchers to guide the formation and maturation of the vascular barrier, offering a powerful tool for studying the pathophysiology of iBRB and developing new therapies for retinal diseases such as DR [128] (Table 2).

## 6. Conclusions

Experimental models for investigating the pathophysiology of DR comprehend the use of different species of animals because of the enormous difficulties in recreating the human retinal microenvironment *in vitro*. Inevitably, the responses of non-human retinas to hyperglycemic conditions are different among species due to the different genetics and habits of animals. Furthermore, a large amount of literature highlights a lack of reproducibility, depending on what species is used for investigating innovative therapeutic options for the treatment of DR. In addition, after the recognition of 3R principles, ethical concerns in using genetically modified animal models are arising and pushing researchers to develop “alternative” models. In particular, due to the involvement of different types of cells that orchestrate the progression of glucose damage in neuronal, glial, and vascular cells, innovative and complex *in vitro* methods have been developed in the last few years. The primary advantage in terms of the reliability of the response is the use of human-derived cells, which avoid the genetic differences among species. Furthermore, the knowledge of developing reconstructed tissues paves the way not only for studying the effectiveness of new molecules in restoring retinal health but also for presenting a possible strategy for clinical practice. Indeed, the *in vitro* construction of retinal tissue could be an extremely innovative strategy for the treatment of those pathologies characterized by progressive and inevitable visual impairment with possible transplantation procedures. Although huge steps forward have been taken in developing complex *in vitro* models, there are still gaps in their reliability, both in their response to harmful stimuli, including HG, and in their response to innovative potential therapeutic options. Furthermore, HG is probably one of the main conditions reported in the literature to induce diabetic-like damage, but it has some limitations. Diabetes is characterized by systemic hyperglycemia, and this state promotes the activation of secondary pathways for the metabolism of certain amounts of glucose (for example, polyol pathway, AGEs accumulation, and PKC activation) in the retinal tissue and in the whole vascular tree. Exposing retinal cells to high glucose levels does not reflect the different types of circulating mediators. Other methods exist for recreating *in vitro* diabetic conditions: one method is the use of methylglyoxal (one of the major precursors of AGEs), but it presents some limitations. In general, AGEs need an overexpression of the receptor RAGE, and the *in vitro* treatment with methylglyoxal is not able to induce such a condition in a limited period of time.

Thus, great effort is still needed in standardizing *in vitro* tests that can be potentially used to evaluate personalised drug delivery, speed up drug development, and reduce the need for animal testing.

To further enhance the comprehension of the disease, the researchers employed the construction of organoids using co-cultured neural retina and RPE explants, thereby facilitating more efficacious interactions between photoreceptors and RPE cells. The most advanced *in vitro* technique is the organ-on-a-chip, which can replicate the functions and behaviors of human organs or tissues on a small scale. These devices provide a more accurate model for studying human biology, disease, and drug testing compared to traditional 2D cell cultures or animal models because they can replicate complex tissue structures, such as cell-to-cell interactions, organ-specific functions, and responses to external stimuli like drugs or toxins. Furthermore, they can be used for drug development in the extremely promising field of personalized medicine. Indeed, the organ-on-a-chip can be exploited to culture patient-derived cells, creating personalized models for testing how a specific individual’s cells will respond to different treatments.

However, many challenges need to be faced. The overall complexity of organ-on-a-chip systems, which, although more accurate than simpler cell cultures, still do not fully capture the complexity of an entire organism. The standardization procedure needs to follow a specific protocol. To date, each chip design is unique, which makes comparing results across studies difficult.

Finally, many organ-on-chips are still in the early stages of development and scaling them for large-scale drug testing or personalized medicine remains a work in progress.

## Figures and Tables

**Figure 1 cells-13-01864-f001:**
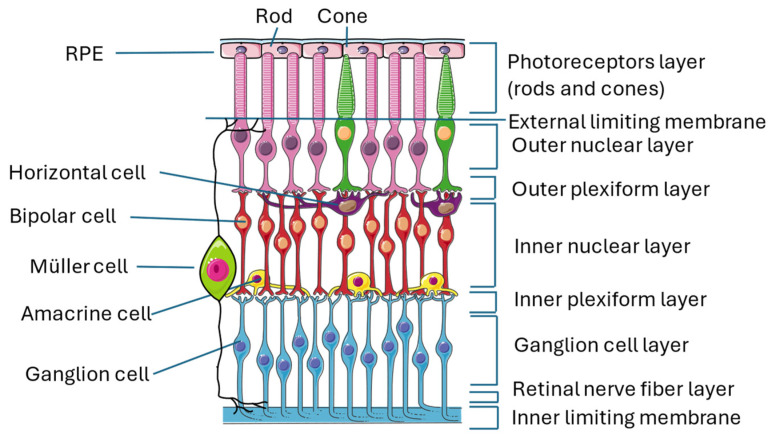
Structure of the retina. The image illustrates the multiple layers that constitute the retina and the numerous interconnections among the various cell types, thereby exemplifying the intricate complexity of this organ. This figure was created through https://smart.servier.com/ (accessed on 15 September 2024). All the resources are freely available for use under CC BY 4.0.

**Figure 2 cells-13-01864-f002:**
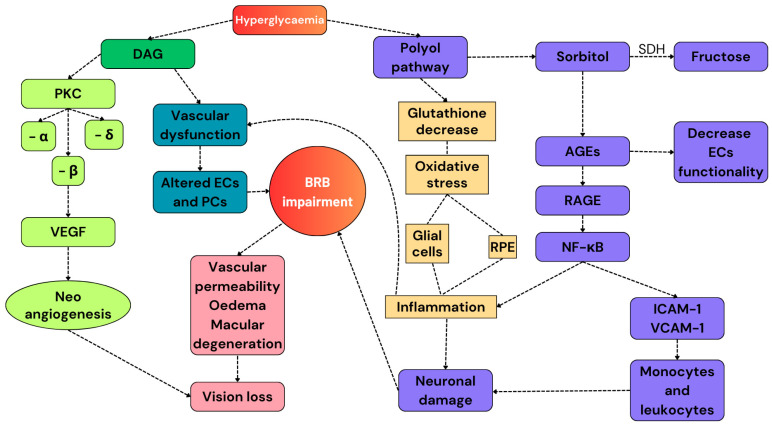
A summary of the pathways involved in the development of DR. Hyperglycemia activates the secondary pathways. The polyol pathway reduces the number of endogenous antioxidant molecules and increases the expression of RAGEs, resulting in the exacerbation of the pro-inflammatory response. Diallyl glycerol activates PKC, whose subunit β promotes the release of VEGF and promotes neoangiogenesis. All these alterations cause BRB impairment and, finally, vision loss. DAG: diallyl glycerol; PKC: protein kinase C; VEGF: vascular endothelial growth factor; ECs: endothelial cells; PCs: pericytes; RPE: retinal pigmented epithelium; AGEs: advanced glycation products; and SDH: sorbitol dehydrogenase.

**Figure 3 cells-13-01864-f003:**
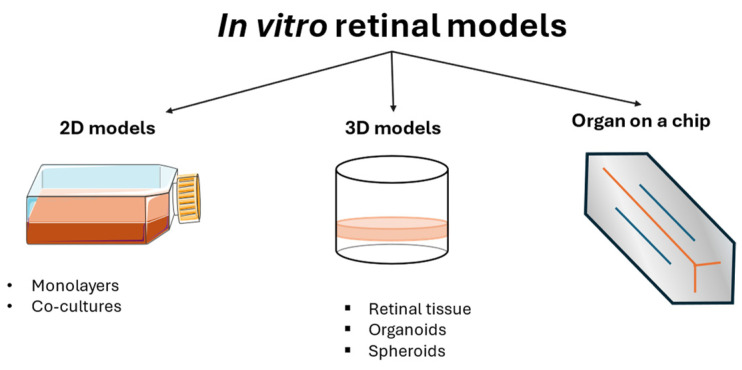
Schematic overview of *in vitro* models of DR.

**Table 1 cells-13-01864-t001:** Cellular models.

Cell Line	Species	2D Models	References
MCs *	Mouse/human	HG * conditions, studies on oxidative stress, and RAGE * regulation	[83]
BV-2 *	Mouse microglia	Role of hypoxia in pathological mechanisms	[84,85]
HREC * and hRPs *	Human	Studies on BRB * function	[86,87]
ASCs *	Mouse/human	Differentiation toward PCs *	[88,89,90,91,92,93]
HMC-3 * and PUMC *-HUVEC-T1 *	Human	Effect of HG * on vascular development	[94]
ARPE-19 * and HUVEC *	Human	Development of BRB *	[95]
Müller and microglia	Mouse	Study of the effect of activated MCs in an inflammatory state	[96]
HESC *	Human	Differentiation toward PCs *	[97]

* ASCs = adipose stem cells; hESC = human embrionic stem cell; hiPSC = human induced pluripotent stem cells; HREC = human retinal endothelial cells; hRPs = human retinal pericytes; HMC-3 = human microglia C3 cells; PUMC-HUVEC-T1 = Peking Union Medical College-human umbilical vein ECs T1; ARPE-19 = spontaneously arising retinal pigment epithelial cells; RAGE = receptor for advanced glycation end products; PCs = pericytes; HG = high glucose; BRB = blood retinal barrier; and MCs = Müller cells.

**Table 2 cells-13-01864-t002:** 3D *in vitro* models and organ-on-a-chip models.

Type	Species	3D Models	Use	References
Retinal tissue	Mouse (RPE *) Human (hESC *)	Retinal tissue	Drug screening and regenerative medicine	[117]
	hiPSCs	Retinal cups	Drug screening and regenerative medicine	[118]
Organoids	Human (hiPSC *)	Two synaptic and three nuclear layers of the retina	Studying cellular targets	[119]
	Human (hiPSC */hESC-H9 *)	Capillary networks	Diabetes vasculopathy	[120]
	Human (hiPSC *)	Retinal monolayer with photoreceptors	Drug testing, disease modeling	[121]
	Human (hiPSC *)	3D *in vitro* model of DR	Studies on the mechanisms of DRN *	[122]
Spheroids	Human (hRPE *)	RPE spheroids	RPE, Bruch’s membrane and lipoproteins studies	[123]
oBRB *	Human (HUVEC */NHLF */ARPE-19 *)	oBRB-choriocapillary model	Mimic the architecture and function of the oBRB	[124]
Organ-on-a-chip	Human (hiPSC *-derived RPE)	Engineered oBRB *	BRB * studies	[125]
	Human (ARPE-19 */SH-SY5Y */HREC *)		Development of a device of iBRB *, neuroretina, and oBRB *	[126]
	Human (hiPSC *)	Retinal model with vasculature-like perfusion	Drug testing, studies on the interaction of photoreceptors and RPE *	[127]
	Human (hRMVECs */hRPs */hRAs *)	iBRB	Studies on barrier properties for developing new therapies for retinal diseases	[128]

* RPE = retinal pigmented epithelium; hESC = human embrionic stem cells; hiPSC = human induced pluripotent stem cells; hRPE = retinal pigment epithelial cells; DR = diabetic retinopathy; DRN = diabetic retinal neurodegeneration; HUVEC = human umbelical vein endothelial cells; NHLF = normal human lung fibroblasts; ARPE-19 = spontaneously arising human retinal pigmented epithelial cells; SH-SY5Y = neuroblastoma cell line SK-N-SH; HREC = human retinal ECs; iBRB = inner blood retinal barrier; oBRB = outer blood retinal barrier; hRMVECs = human retinal microvascular endothelial cells; hRPS = human retinal pericytes; and hRAs = human retinal astrocytes.

## Data Availability

Not applicable.

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
