# Peer review of "In Vitro Models of Diabetes: Focus on Diabetic Retinopathy"

_cells, 2024, doi:10.3390/cells13221864_

Round 1
Reviewer 1 Report
Comments and Suggestions for Authors
Galgani and cowoerkers present a review on “Diabetic Retinopathy In Vitro Models: Where are We Now?” Although this could be an interesting review of this specific field, considerable revision is needed in order to have an updated and more accurate manuscript adding novelty to other reviews recently published.
Major comments:
1. The title does not reflect the content of the review and needs adjustment. The review is more about potential in vitro models to be used as diabetic in vitro models, as for major part of them there are few or none examples where these models were already applied for DR.
2. A carefully revision of the literature and of the used references is needed. Some references are not updated. For example, in line 259-260, since 2010 (ref. 59) a high number of differentiation protocols have been developed. Important publications describing new models of DR were not discussed, such as 10.1038/s41467-024-45456-z, 10.1007/s44164-024-00068-1, 10.1021/acsbiomaterials.3c00326
3. Figure 1 clearly was obtained from BioRender. Authors need to follow BioRender rules and mention it in the figure legend. Also, authors need to present BioRender publication licence for the corresponding figures.
4. Schemes 1 and 2 are too simple, and legends need to be more detailed, or need to be removed as they do not bring any extra information to the text. Figure 2 also is to simple and could be improved.
5. Lines 279 to 296. Ref. 64 and review 65 are from 2012 and 2014. Clarify in the text if there are examples of applications of these models on DR studies since then.
6. Line 351. “scientists have recently shifted from using 2D to 3D tests”: this is not accurate. There is a tendence to explore 3D models but these are still offering challenges on reproducibility, require specific training and equipment. Also, 2D models still offer advantage on high-throughput screenings. The tendance is to use the different models in a pipeline from less complex to more complex systems.
7. Line 354; Section 4.2.1. Two examples are presented of models published some years ago: ref. 70 and 71. Where these models used by the same authors and by others since then to study DR or other diseases?
8. Line 483. “…since can be used for transplantation”. Please rephrase it as this is not possible yet although many studies are exploring it for future applications.
9. It will be interesting to explore how the DR models are being induce, besides high glucose concentration. Which other molecules are being used in different studies to model DR? That is the opinion of the authors the limitations of this models to reproduce the DR microenvironment?
Minor comments:
Figures, schemes, and tables are not mentioned in the main text.
Abbreviations use needs revision (p.e. diabetic retinopathy by DR)
In 4.1.2, lines 302 and 313, cell lines names should be replaced by the cell types as in line 329.
Line 453, this sub title could be removed as is the single one in the section 5. Update of this section with recent publications is needed.
Reviewer 2 Report
Comments and Suggestions for Authors
This manuscript can be improved with minor revisions.
The section 3, on Development of DR can be significantly revised to clarify the pathophysiology, and the roles of each of the contributory cells. The differences between Schemes 1 and 2 are not currently very clear to the reader. These schema should be appropriately revised.
The contribution of individual cell models need expansion. What's the role of EC, pericyte, and Muller cell models in vitro?
Future directions need further input.
Reviewer 3 Report
Comments and Suggestions for Authors
Within this review, Galgani et al. aim to give an update on the state of the art of diabetic retinopathy, focusing on the in vitro models used to study this disease. The review starts by introducing diabetic retinopathy and explaining its main causes, features and cell types involved. A brief overview is then given on the pros and cons of using animal models. In the main text, the authors describe in detail the structure of the neuroretina and its vascular network to elaborate on the different stages of the disease and the signalling pathways involved. The rest of the manuscript is dedicated to the in vitro models currently exploited to study diabetic retinopathy with a focus on the latest technologies.
Overall, the review is well-written and easy to follow. All the major aspects of diabetic retinopathy and the in vitro models currently in use have been properly addressed.
It’s not clear what the authors mean by pericycle.
I think the role of pericytes has been neglected and this cell type should be mentioned as it covers an important role in the BRB. Co-cultures of endothelial cells and pericytes should be discussed as they represent important models of the iBRB.
The paragraph on Retinal Tissue should be elaborated more and additional details about the study cited should be mentioned.
Besides Figure 1, which is very good, all the other figures are a bit poor and don’t add much to the manuscript. Details should be added to make the figures more significant and showcase the different signalling pathways and models described.
Round 2
Reviewer 1 Report
Comments and Suggestions for Authors
The authors addressed the major concerns and suggestions raised by the first revision.
The review article quality was improved.